# Identification of drug combinations on the basis of machine learning to maximize anti-aging effects

Sun Kyung Kim[1], Peter C. Goughnour[1], Eui Jin Lee[1], Myeong Hyun Kim[2], Hee Jin Chae[2], Gwang Yeul Yun[2], Yi Rang Kim[2,3]*, Jin Woo Choi[1,4]*

**1** College of Pharmacy, Kyung Hee University, Seoul, Republic of Korea, **2** Center for Research and Development, Oncocross Ltd., Seoul, Republic of Korea, **3** Department of Hematology/Oncology, Yuseong Sun Hospital, Daejeon, Republic of Korea, **4** Department of Life and Nano-pharmaceutical Sciences, Kyung Hee University, Seoul, Republic of Korea

* 99yirang@gmail.com (YRK); jinwoo.ch@khu.ac.kr (JWC)

## Abstract

Aging is a multifactorial process that involves numerous genetic changes, so identifying anti-aging agents is quite challenging. Age-associated genetic factors must be better understood to search appropriately for anti-aging agents. We utilized an aging-related gene expression pattern-trained machine learning system that can implement reversible changes in aging by linking combinatory drugs. *In silico* gene expression pattern-based drug repositioning strategies, such as connectivity map, have been developed as a method for unique drug discovery. However, these strategies have limitations such as lists that differ for input and drug-inducing genes or constraints to compare experimental cell lines to target diseases. To address this issue and improve the prediction success rate, we modified the original version of expression profiles with a stepwise-filtered method. We utilized a machine learning system called deep-neural network (DNN). Here we report that combinational drug pairs using differential expressed genes (DEG) had a more enhanced anti-aging effect compared with single independent treatments on leukemia cells. This study shows potential drug combinations to retard the effects of aging with higher efficacy using innovative machine learning techniques.

## Introduction

Aging is recognized as a direct or indirect cause of many diseases [1]. The suppression and reversion of aging is a considered methodology to prevent and cure its related diseases [2]. Aging is a complex genetic phenomenon, and the accepted strategy to delay aging is to regulate gene expression [3] rather than focusing on developmental mutations [4, 5].

However, due to the complex networks that link aging-related factors, computation and machine learning using previously accumulated data may be a promising approach to understand hidden causative gene expression patterns and identify new anti-aging drugs [6].

**Funding:** This research was supported by a grant from the National R&D Program for Cancer Control, Ministry of Health and Welfare, Republic of Korea to YRK and JWC (HA17C0039), and the Basic Science Research Program through the National Research Foundation of Korea (NRF-2017R1A5A2014768) funded by the Ministry of Education, Republic of Korea (2017R1C1B5017615). Oncocross Ltd. provided support for this study in the form of salaries for MHK, HJC, GYY, and YRK. The specific roles of these authors are articulated in the 'author contributions' section. The funders had no role in study design, data collection and analysis, decision to publish, or preparation of the manuscript.

**Competing interests:** The authors have read the journal's policy and the authors of this manuscript have the following competing interests: MHK, HJC, GYY, and YRK are paid employees of Oncocross Ltd. There are no patents, products in development or marketing products to declare. This does not alter our adherence to PLOS ONE policies on sharing data and materials.

Using computational and bioinformatics strategies, researchers can now generate and analyze different kinds of data and drug repositioning (repurposing); in particular, finding new uses for existing drugs has become a popular method for unique drug discovery [7, 8]. These advances in drug screening are primarily due to the increasing number of gene expression-profiling analyses that have exhibited desired drug effects.

Recent collaborative efforts have combined many different fields of study, and the computer-aided drug discovery/design [9] method is adopted to facilitate the time-consuming process and to increase the effectiveness of drug discovery [8]. Recently, many databases for *in silico* drug development have been established; for example, Drugbank (2006) is a drug database with comprehensive drug target data [10], and PDTD (2008) is a web-accessible protein database for drug target identification [11]. Furthermore, large public databases are available that contain information regarding relationships between drugs and genes, which have made drug repositioning or repurposing more accessible.

CMap (Connectivity Map), which can reveal unexpected connections among drugs, genes, and diseases [12], is a representative example of the use of drug repositioning. CMap shows relationships between different biological states through drug treatment using gene expression profiles and signatures [13]. Thus previously developed compounds can be predicted and applied for non-targeted diseases. CMap has been applied to various drug development processes. It is used as a logical reference because each chemical induces different gene expressions, and all illnesses are caused or accompanied by changes in their gene expression.

Nevertheless, CMap has limitations, such as the use of a database with different cell lines for gene expression profiles [14] that does reflect the real impact of a drug on a particular cell line [15]. CMap is limited to only five cell lines such as: MCF7 (Breast cancer), PC3 (Prostate cancer), HL60 (Leukemia), ssMCF7 (Breast cancer-charcoal-stripped serum), SKMEL5 (Skin cancer). Therefore, if there are some diseases that we want to target that are not in these five cell lines, it is difficult to derive an accurate drug lists. Also, Cmap can be used to select drugs that can down-regulate the up-regulated genes and vice versa. However, the drug's gene expression pattern (derived from Cmap) and the inserted gene expression pattern used in the initial analysis do not always match. In other words, the drug derived from CMap does not cover all of the gene expressions for that particular disease.

These above shortcomings may reduce the success rate of identifying potent drug repositioning. Additionally, it is difficult to alter the total gene expression pattern in the expected direction using a single drug. Thus, similar to drug combination therapy, wherein multiple chemicals are combined to treat disease [16], by combining two to three compounds with different mechanisms of action, researchers can overcome shortcomings in the drug repositioning process. Drug combinations can also reduce the required concentration of individual drugs [17].

This paper discusses how drug combinations can broadly manage a disease mechanism that can be used to find unique drug targets. Also, it shows how machine learning can assist in combining previous drugs that may have been missed by prior mathematical algorithms. In the current study, gene expression profiles for acute myeloid leukemia patients were downloaded from Gene Expression Omnibus (GEO), a database repository of high throughput gene expression data and hybridization arrays, chips, microarrays. The purpose of our model was to determine what kind of gene, especially in PBMC (peripheral blood mononuclear cell), was differentially expressed between two groups the young and the old. But as any normal PBMC cohort didn't meet the enough number of population, we used GSE6891 which was collected for leukemia research including the 'age' information.

After that, differentially expressed genes (DEGs) in the aged group were determined with an artificial neural network [18] and matched with drug-induced gene expression profiles

from CMap to predict drug candidates that can reverse the DEG pattern of the aged group. Using a deep neural network (DNN), an artificial neural network (ANN) with multiple layers between the input and output layers was used to find the correct mathematical manipulation to turn the input into the output, whether it be a linear relationship or a non-linear relationship. An artificial neural network is an interconnected group of nodes, inspired by a simplification of neurons in a brain. A two-stage search strategy was adopted to find drug combinations to reverse the aging-related target DEGs more effectively than the one-drug approach.

The purpose of our DNN model was to determine if a specific gene was differentially expressed between two samples. In many studies, fold-change approach and statistical methods including t-test and non-parametric test has been used as criterion to select differentially expressed genes (DEGs). But these approaches are sensitive to outliers or sample size. In this study, our goal was to develop a DEGs selection method which is robust to the sample size. The training dataset consists of the DEGs of the results of LIMMA from multiple GSEs with various sample size to this end.

## Materials and methods

### Deep neural network model

In this study, we built a deep neural network (DNN) model using the Tensorflow framework [19] to predict whether a specific gene was differentially expressed between two samples. The model is composed of an input layer of two nodes, three hidden layers with ten nodes each, and an output layer with two nodes that represent "up-regulation" and "down-regulation." Rectified linear unit (ReLu) and Softmax was used as an activation function of hidden layers and output layer, respectively. Gene expression data were downloaded from the Gene Expression Omnibus (GEO) site (https://www.ncbi.nlm.nih.gov/geo). We selected 13 datasets of leukemia samples to train our DNN model. The training dataset consists of 730 samples, with 108 and 622 samples in the control and disease groups, respectively. We used the *limma* package in GEO2R [20] (https://www.ncbi.nlm.nih.gov/geo/geo2r/) to compute $p$ value for each probe with a moderated t-test. The null hypothesis was that the gene was expressed the same in the disease group and the control group. We cut off those probes with $p$ value > 0.1. We labeled the probes as "up" or "down" according to the expression values. Then we trained the DNN to predict whether a specific gene is differentially expressed between two samples. According to the experimental design of each dataset, we trained the model with expression value pairs of each gene between samples, one from the control group and the other from the disease group within individual datasets.

### Gene expression data of aged group

Microarray data were downloaded from NCBI GEO with accession number GSE6891, which were derived from 461 acute myeloid leukemia patients [21, 22]. GSE6891 is the clinical data that represents the gene expression profiles of AML samples of two independent cohorts (n = 247 and n = 214). Data analyses were carried out to discover and predict prognostically relevant subtypes in AML (<60 years) based on their gene expression signatures. We categorized these data into two groups based on the patients' age. Samples from patients older than 50 were included in the aged group, and those younger than 30 were included in the normal (young) group. The number of samples in the normal and aged groups were 84 and 159, respectively. Differentially expressed genes were identified with GEO2R in the same way as described in the 'Deep neural network model' section above. Lastly, we mapped the probe IDs to Entrez-gene IDs.

## Drug-induced gene expression data

CMap is a resource that uses transcriptional expression data to probe relationships between diseases, cell physiology, and therapeutics [23]. The gene expression data is downloaded from CMap, which shows transcriptomic changes following drug treatment. In this study, we analyzed gene expression data from 1,072 drug experiments with HL-60 cells.

## Cell culture

HL-60 cells were cultured at $1 \times 10^5$ cells/mL in Roswell Park Memorial Institute (RPMI) 1640 medium supplemented with 10% fetal bovine serum (FBS) and 1% penicillin/streptomycin at 37 ˚C in a 5% $CO_2$ humidified atmosphere.

## Cell viability analysis

Trypan blue staining was performed to assess cell viability. HL-60 cells were incubated in 48-well plates at $3.0 \times 10^4$ cells per well. After chemical treatment with or without pre-treatment with 10μM hydrogen peroxide, cells were diluted by trypan blue working solution and counted with a cell counter to allow for growth curve construction [24].

## Reactive oxygen species (ROS) assay

HL-60 cells were seeded in 96-well plates at a density of $3.0 \times 10^4$ cells per well and then incubated with 10μM hydrogen peroxide to achieve ROS-induced senescence. After 4hr, each reagent was treated for 36hr in serum-free conditions. For cells, ROS production was measured with the ROS Detection Assay Kit (Biovision, Inc., Milpitas, CA, USA) according to the manufacturer's protocol [25]. The plate was finally scanned with the CLARIOstar® Plus microplate reader (BMG Labtech, Ortenberg, Germany). The drugs were administered at the following concentrations: 30nM trichostatin A, 1μM vorinostat, 5μM anisomycin, 2mM metformin, 20μM danazol, 10μM glibenclamide, 20μM ampyrone, and 20μM chlorzoxazone.

## Nuclear morphology assay

For fluorescence microscope detection of nuclear morphology changes, chemically treated HL-60 cells were washed twice with phosphate-buffered saline (PBS) and fixed with 4% para-formaldehyde in PBS for 15 min. For nuclear staining, cells were incubated with 1.0μg/mL 4´,6-diamidino-2-phenylindole (DAPI). Fluorescent microscopy images were obtained using a fluorescence microscope system [26].

## Senescence-associated beta-galactosidase staining

HL-60 cells were seeded in 24-well plates at a density of $3.0 \times 10^4$ cells per well and then incubated with 10μM hydrogen peroxide to achieve senescence. After 4hr, each reagent was treated for 36hr in serum-free conditions. For cells, β-galactosidase assays were performed using a Biovison Beta-Galactosidase Staining Kit (Biovision, Inc., Milpitas, CA, USA). The protocol was conducted according to the manufacturer's instructions [27]. After fluorescent micros-copy images were obtained using a fluorescence microscope system, beta-galactosidase positive cells were counted with a cell counter.

## Results

### Production of the gene expression pattern of the aged group and drug response patterns

The GEO2R program [20] (https://www.ncbi.nlm.nih.gov/geo/geo2r/) was used to calculate differentially expressed genes between the aged and normal groups of leukemia patients cohort (GSE6891), which 1,415 genes had a $p < 0.1$. The samples from patients older than 50 were included in the aged group, and those younger than 30 were included in the normal (young) group, as shown in S1 Fig. Using our deep neural network (DNN), we computed the predictive value of each probe and ruled out the probes with their predictive value less than 0.95 (Fig 1A). The model architecture and other parameters are shown in S1 Table. We selected 13 datasets of leukemia samples to train our DNN model (S2 Table). The performance measured by the loss, accuracy, and area under the curve [28] is shown in S3 Table. Then we classified the probes as "up-regulated" or "down-regulated" according to its predictive value (S2 Fig). After that, we substituted the probe IDs with Entrez IDs to match them with the drug-induced gene expression data. Our approach resulted in the DEGs list consisting of 1,293 probes (Fig 1B), in which blue represents up-regulated genes, and red represents down-regulated genes in the aged group compared to the normal group. In this study, which represents the original DEGs (o-DEG) consisted of 676 up-regulated genes and 617 down-regulated genes. Drug-induced DEG (d-DEG) pattern was created similarly. In the drug-induced expression data, DEGs were selected with limma ($p$ value $\leq 0.1$) and filtered with our DNN (predictive value $\geq 0.95$) (Table 1, and S4 Table).

### Initial and second matching

To identify a DEG pattern related to therapeutic direction, initial matching was conducted, which reversed the o-DEG pattern, so that up-regulated genes were down-regulated and vice versa. We called this reversed DEG pattern "i-DEG," which means the "ideal" state to reverse the current o-DEG of the aging state. We screened common genes between the i-DEG pattern and each d-DEG pattern accordingly and built a common DEG (c-DEG) pattern called initial matching (Fig 2A and 2B). The gene number in the c-DEG of each drug was scored, and chemicals were ranked based on score (S5 Table). Vorinostat [29], trichostatin A [30], lycorine, and anisomycin were designated as results of the initial matching, but lycorine was ruled out due to its expected toxicity (Table 2).

Since the drugs found in initial matching cannot reverse all the gene expressions in the o-DEG pattern, a second matching phase was performed to discover other combinatorial drugs to maximize the number of reversed genes with drugs found in initial matching. In the second phase, the genes covered by the chemicals found in initial matching were excluded when scoring the common genes between the i-DEG list and d-DEG patterns. The suggested drugs, in combination with the drugs identified in initial matching, are shown in Table 3 and S6 Table. The o-DEG and d-DEG patterns of vorinostat were compared, and the reversed match pixels were removed and expressed as white color. Thus, white color represents higher simulated efficacy of the drug. Combinatory treatment with anisomycin was predicted by broader white areas and was expected to reverse more o-DEGs in combination with vorinostat (Fig 3). We performed analysis and found genes that were both up-regulated in the first drug and down-regulated in the second drug screening, and vice versa. A total of 14 drug combinations, genes that overlap with Vorinostat and Anisomycin were only valid. Genes down-regulated in Vorinostat were up-regulated with effective values in Anisomycin (logFC>1, p<0.1). However, out of a total of 18,476 genes (down-regulated in Vorinostat and up-regulated in Anisomycin),

**A**

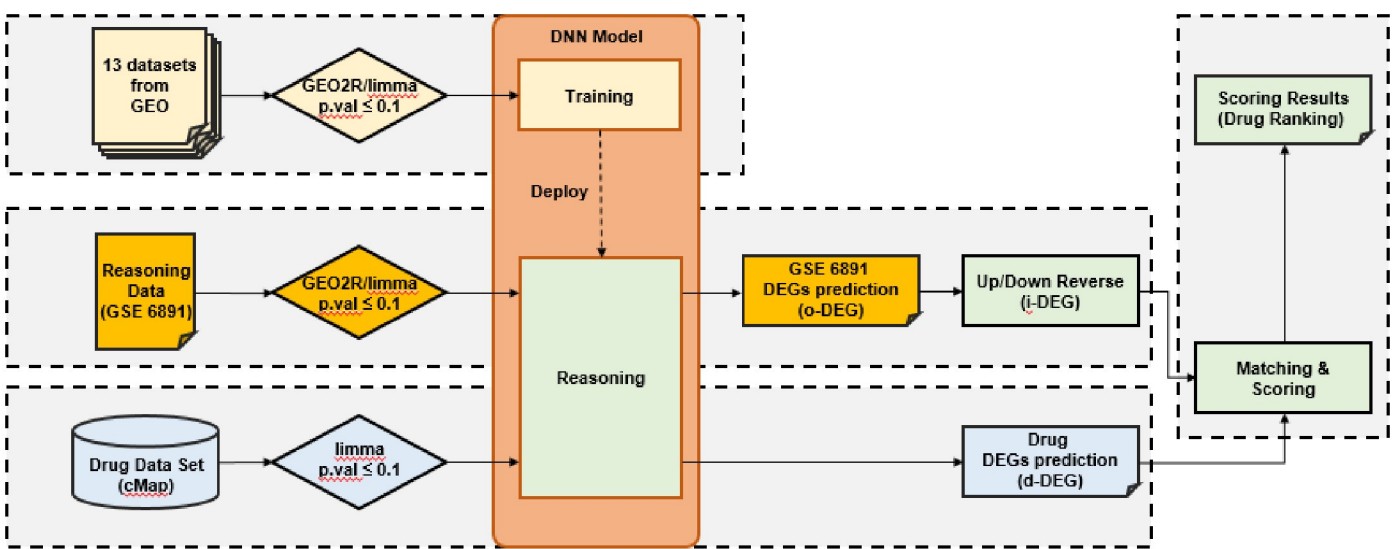

**B**

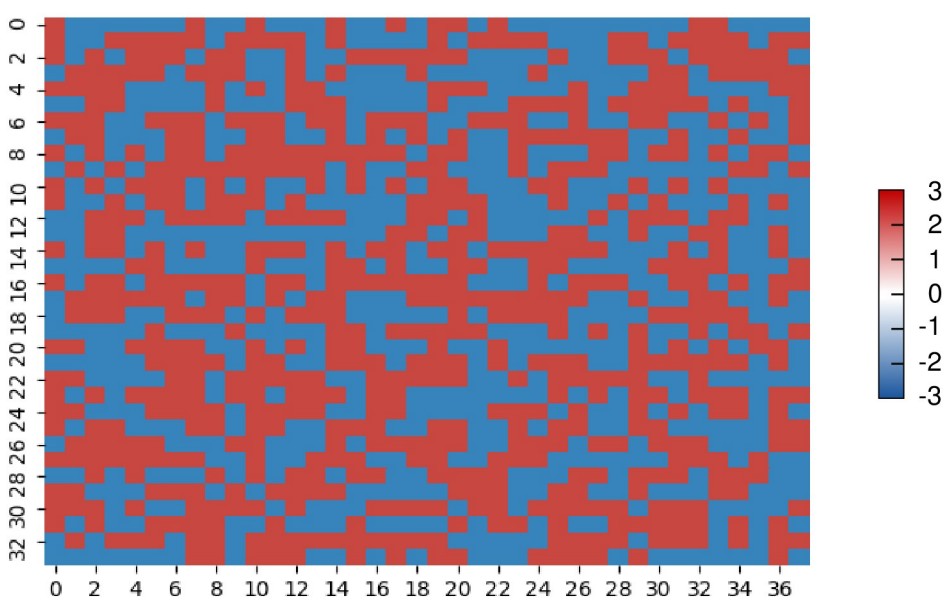

DEG list consisting of 1,293 genes

**Fig 1. Original differentially expressed genes (o-DEGs) between normal and aged groups. (A)** A schematic of training deep neural network (DNN) model with gene expression data from 13 datasets from GEO. DEGs pattern of an aged group (o-DEG) was built and transformed into i-DEG for matching with d-DEG. d-DEG was created with drug-induced gene expression data from the Connectivity Map. i-DEG and d-DEG are matched and scored to rank drugs. **(B)** A heatmap of up-regulated genes (marked in blue) and down-regulated genes (marked in red) in the aged group compared to the normal group. The total gene number in this pattern is 1,293.

**Table 1. Differential gene expression patterns using DNNs.**

| Symbol | Score | Down | Up | Symbol | Score | Down | Up |
|---|---|---|---|---|---|---|---|
| CD34 | 99.9991 | 0 | 1 | PPBP | 0 | 1 | 0 |
| TRH | 99.9976 | 0 | 1 | CA2 | 0 | 1 | 0 |
| HPGDS | 99.9962 | 0 | 1 | HOXA5 | 0 | 1 | 0 |
| LOC100505501 | 99.9947 | 0 | 1 | HOXA9 | 0 | 1 | 0 |
| C1QTNF4 | 99.9941 | 0 | 1 | HOXA10 | 0 | 1 | 0 |
| CYTL1 | 99.9903 | 0 | 1 | HBD | 0 | 1 | 0 |
| MSLN | 99.9884 | 0 | 1 | COL4A5 | 0 | 1 | 0 |
| ITM2A | 99.9883 | 0 | 1 | BEX3 | 0 | 1 | 0 |
| JUP | 99.988 | 0 | 1 | PF4 | 0 | 1 | 0 |
| POU4F1 | 99.9846 | 0 | 1 | EREG | 0 | 1 | 0 |
| CD200 | 99.9808 | 0 | 1 | HOXB2 | 0 | 1 | 0 |
| MN1 | 99.9803 | 0 | 1 | HBM | 0 | 1 | 0 |
| TSPAN7 | 99.979 | 0 | 1 | EPB42 | 0 | 1 | 0 |
| UMODL1 | 99.979 | 0 | 1 | XK | 0 | 1 | 0 |
| EGFL7 | 99.9787 | 0 | 1 | HOXB3 | 0 | 1 | 0 |
| KIF4CP | 99.9749 | 0 | 1 | RTL8C | 0 | 1 | 0 |
| TPSB2 | 99.9734 | 0 | 1 | PROK2 | 0 | 1 | 0 |
| BAALC | 99.9709 | 0 | 1 | NKX2-3 | 0 | 1 | 0 |
| PLTP | 99.969 | 0 | 1 | SLC40A1 | 0 | 1 | 0 |
| DNTT | 99.968 | 0 | 1 | TMEM176A | 0 | 1 | 0 |
| UHRF1 | 99.9659 | 0 | 1 | PRKAR2B | 0 | 1 | 0 |
| ITGA6 | 99.9647 | 0 | 1 | DMTN | 0 | 1 | 0 |
| PRAME | 99.9627 | 0 | 1 | ALAS2 | 0 | 1 | 0 |
| TPSAB1 | 99.957 | 0 | 1 | MEIS1 | 0 | 1 | 0 |
| ZBTB8A | 99.9552 | 0 | 1 | GYPA | 0 | 1 | 0 |

only two genes IL6 and CYP1A1 were reversely regulated effectively, so it is shown that Anisomycin does not neutralize the effect of Vorinostat (S7 Table).

## Validation of the anti-aging effect via ROS staining

We evaluated drug cytotoxicity with trypan blue assay, and the no observed adverse effects level (NOAEL) was selected as an experimental dose with sub-lethal conditions (S3A–S3H Fig). To rapidly induce cellular senescence, we tried to generate reactive oxygen species (ROS), chemically reactive chemical species containing oxygen, in HL-60 cells with hydrogen peroxide, and then directly measured the ROS concentration. In the first round of drug matching, intracellular ROS levels in trichostatin A-, vorinostat-, and anisomycin-treated cells decreased by 9% on average compared with their levels in the experimental control. ($^*$p $<.05$) Interestingly, their combinatorial treatment slightly and significantly lowered the ROS concentration about 15% more than that with a treat by each chemical alone. ($^*$p $<.05$) The predicted first and second combinatory treatments showed an average efficacy of 27% (Fig 4A and 4B, and S4A Fig). ($^{**}$p $<.01$) Still, the randomly combined treatment did not decrease the ROS concentration significantly (average 14.5%, S4B Fig). Taken together, the combination of drugs seemed to protect against ROS-induced aging.

## Analysis of anti-aging effect based on nuclear shape alteration

Commonly, significant changes in nuclear shape were seen in senescent cells. Untreated cells were all relatively small in size and with adequately regular round morphology. Nevertheless,

**A**

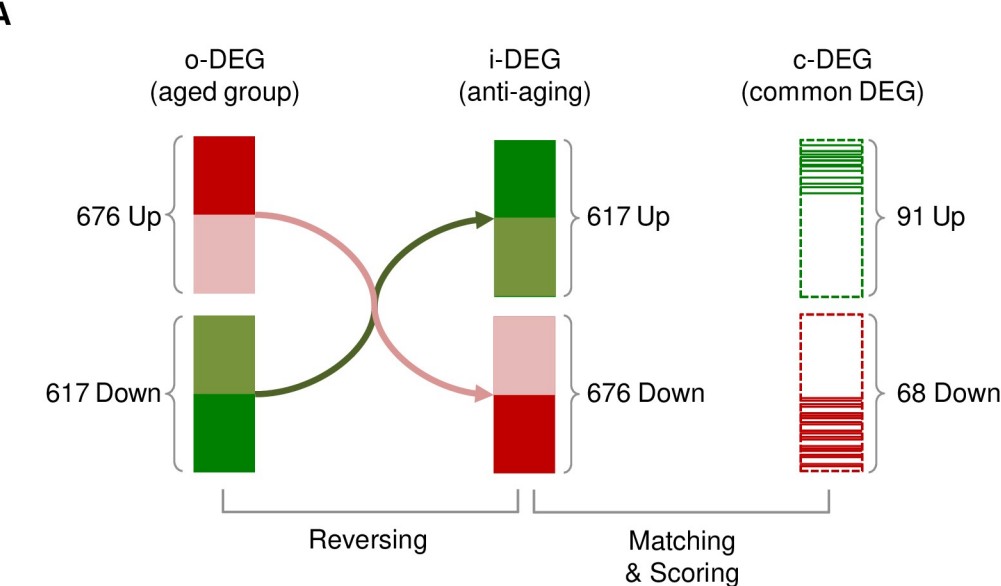

**B**

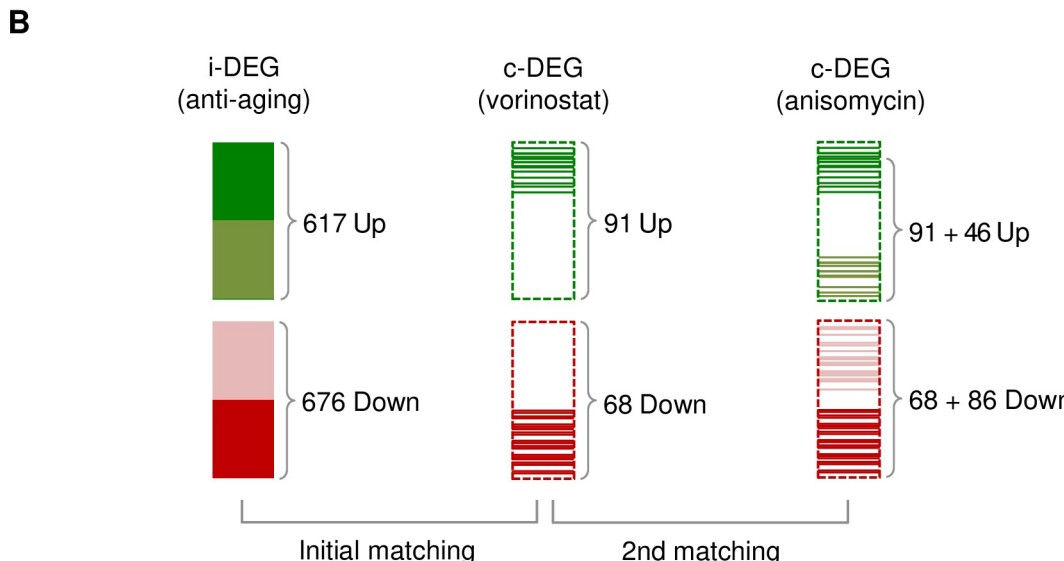

**Fig 2. Flow of initial and second matching. (A)** To identify a differentially expressed gene (DEG) pattern that is expected to have anti-aging effects, we reversed the original DEGs (o-DEGs) of the aged group. The i-DEGs comprised 617 up-regulated genes and 676 down-regulated genes. We performed matching with each drug-induced DEG (d-DEG) to select common DEGs (c-DEGs). In the case of vorinostat, 91 up-regulated genes and 68 down-regulated genes were shared with the i-DEG pattern. **(B)** To maximize the number of reversed genes, second matching was performed. Excluding the drugs chosen in initial matching, drug-induced DEGs (d-DEGs) of each drug were matched with genes not covered by the first drug. In the case of vorinostat and anisomycin, c-DEGs of vorinostat covered 159 genes, and anisomycin, 132 genes.

senescent cells were mainly large and characterized by an irregular distribution of fluorescence [31, 32]. To investigate the morphological alterations that occur in aging nuclei, hydrogen peroxide was used to induce ROS-based senescence in HL-60 cells. The morphology of DAPI-stained nuclei was analyzed (S5 Fig), and the ratio of distorted: altered nuclear shapes were counted. Following treatment with 10μM $H_2O_2$, the first CMap drug treatment restored the

**Table 2. Drugs matched to i-DEG pattern in initial matching.**

| drug | rank | up | down | total |
|---|---|---|---|---|
| lycorine | 1 | 70 | 103 | 173 |
| vorinostat | 2 | 91 | 68 | 159 |
| trichostatin A | 3 | 123 | 32 | 155 |
| anisomycin | 4 | 50 | 101 | 151 |
| emetine | 5 | 64 | 50 | 114 |

nuclear morphology to a more spherical shape by 32% in comparison with the experimental control. ($^*$p <.05) Furthermore, combinatorial drug treatment produced nuclei that were about 25% more circular compared with those treated independently with the first or second round drugs (Fig 5A and 5B, and S6A Fig). ($^*$p <.05) The treatments of glibenclamide in combination with trichostatin A or vorinostat decreased the nuclear shape alteration rate most markedly. Still, the randomly combined treatment did not reduce the nuclear shape alteration significantly. However, an unexpected combinatory treatment of vorinostat with trichostatin A showed an average efficiency of 41% (S6B Fig). ($^*$p <.05) Combination drug treatment restored the nuclear morphology to a spherical shape compared to first drug only treatment.

## Identify of anti-aging effect by senescence-associated beta-galactosidase staining

Senescence-associated beta-galactosidase (SA-β-gal or SABG) is a hypothetical hydrolase enzyme that catalyzes the hydrolysis of beta-galactosidase into monosaccharides only in senescent cells. Senescence associated beta-galactosidase is regarded to be a biomarker of cellular senescence. The alteration of senescent cell numbers was validated with beta-galactosidase staining (S7 Fig). HL-60 cells were co-treated with trichostatin A, vorinostat, anisomycin with combinatorial drug pairs for 36 h after incubation with 10μM $H_2O_2$. The percentage of beta-galactosidase positive cells treated with the first-round CMap drugs (trichostatin A, vorinostat, anisomycin) is slightly decreased compared with their levels in the experimental control (10μM $H_2O_2$ only). The percentage of trichostatin A and danazol or trichostatin A and ampyrone combination-treated cells were decreased 48–51% compared with their levels in the first-round drug-treated cells (Fig 6A and 6B, and S8A Fig). ($^{**}$p <.01) However, the percentage of

**Table 3. Drug combinations suggested in 2nd phase.**

| 1$^{st}$ drug | 2$^{nd}$ Drug | up | down | total |
|---|---|---|---|---|
| vorinostat | Metformin | 73 | 81 | 154 |
| | Anisomycin | 46 | 86 | 132 |
| | Danazol | 42 | 9 | 51 |
| | Glibenclamide | 37 | 14 | 51 |
| | Ampyrone | 26 | 15 | 41 |
| | Chlorzoxazone | 25 | 12 | 37 |
| trichostatin A | Metformin | 39 | 12 | 51 |
| | Danazol | 36 | 10 | 46 |
| | Glibenclamide | 30 | 15 | 45 |
| | Ampyrone | 20 | 15 | 35 |
| anisomycin | Metformin | 107 | 45 | 152 |
| | Trichostatin A | 117 | 26 | 143 |

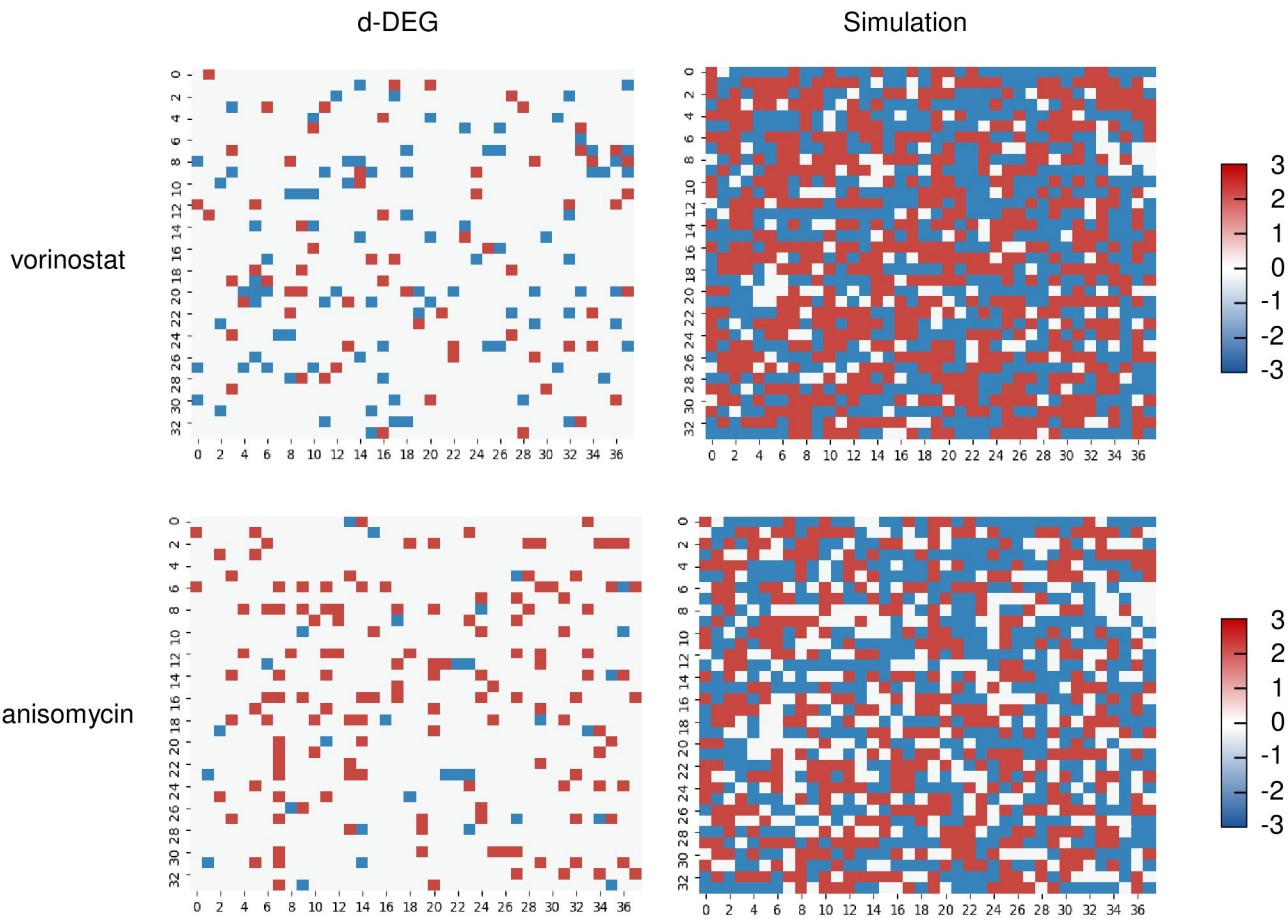

**Fig 3. Simulation of drugs applied to aged samples. Original DEG pattern of the aged group.** The drug-induced DEGs (d-DEGs) of vorinostat and simulated effects when applied to o-DEGs. The d-DEGs of anisomycin and simulated combination effects with vorinostat.

β-galactosidase positive cells treated with randomly combined drug pairs did not decrease significantly (S8B Fig). Taken together, combinational drug treat had a higher percentage of beta-galactosidase compared to single drug.

## Discussion

In this study, we used the public database CMap and focused on devising a new machine learning-combined analytical method to increase the efficiency of target discovery. As expected, the combination of potential drugs displayed better effects than those of single treatment. Here, we suggest how the two-step CMap strategy can be used to improve the success rate of target discovery. We used a modified drug repositioning method that involved the combination of the remaining non-overlapping DEG profile. These DEGs were reinserted into CMap for a second time, and the drugs obtained through the first and second CMap utilization were combined with the expectation of broadening the coverage of existing drugs. Finally, the machine learning strategy was applied to accelerate the process and efficiently solve the input for the third drug.

The three drugs retrieved from the database were trichostatin A, vorinostat, and anisomycin. These were used only as components to evaluate whether 2-step filtered CMap works. Here, we do not insist that these are effective drugs for anti-aging trials; instead, they served as

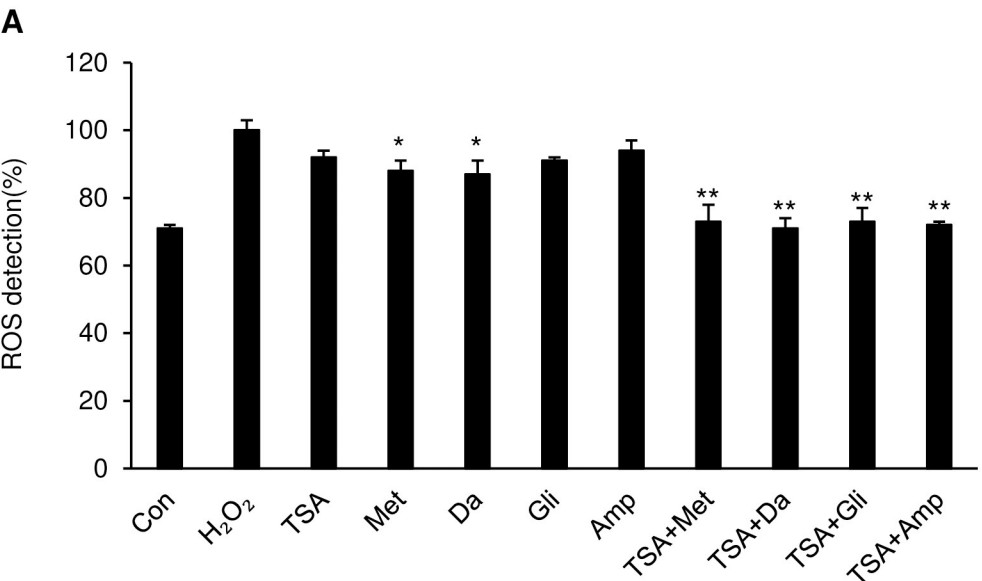

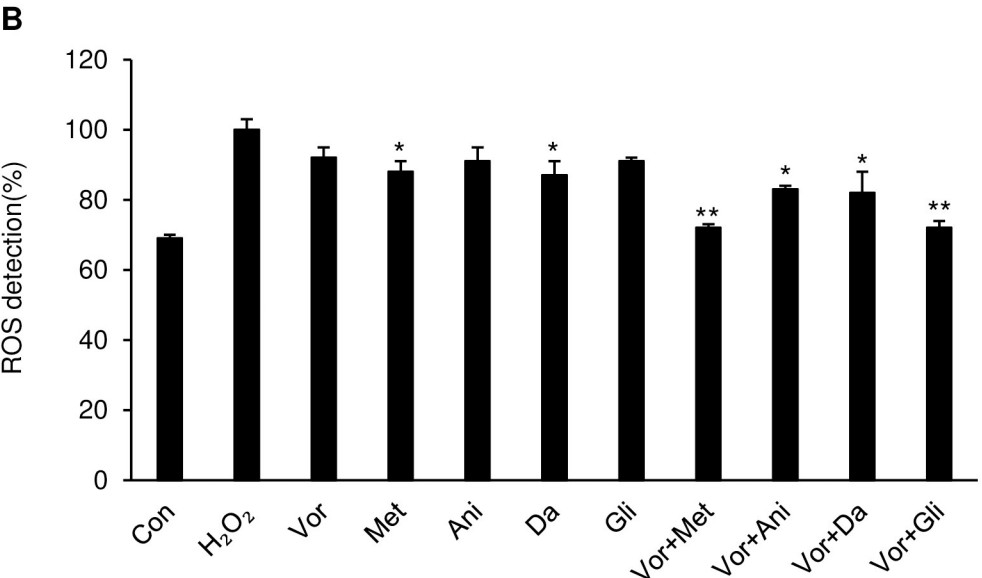

**Fig 4. Removal activity of cellular ROS concentration was detected as an anti-aging effect.** ROS detection measured by fluorescence intensity using a microplate reader. Hydrogen peroxide ($H_2O_2$) was administered to induce ROS-based senescence, then single drugs or combinatorial drug pairs were administered to HL-60 cells to protect against ROS-induced aging. **(A)** HL-60 cells were exposed to $H_2O_2$ (10 µM) for 24 h and then treated with trichostatin A, metformin, danazol, glibenclamide, ampyrone, and each prior drug was co-treated with trichostatin A. or **(B)** with vorinostat, metformin, anisomycin, danazol, glibenclamide and each prior drug with vorinostat for 36 h. The intracellular ROS levels were detected with a microplate reader capable of measuring Ex/Em 495/529 nm spectra and recorded. $^*$p <.05, $^{**}$p <.01 vs $H_2O_2$ group.

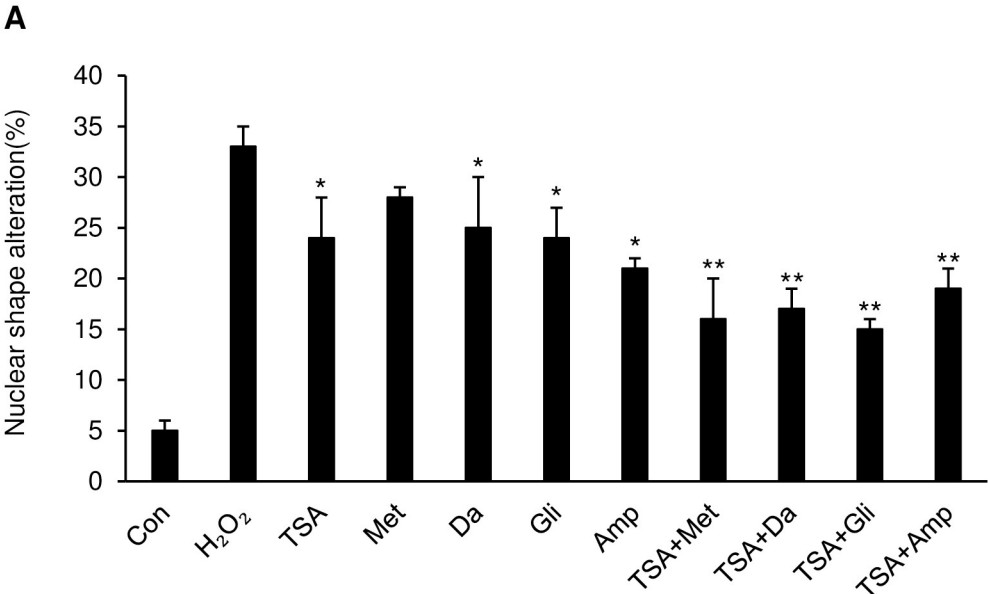

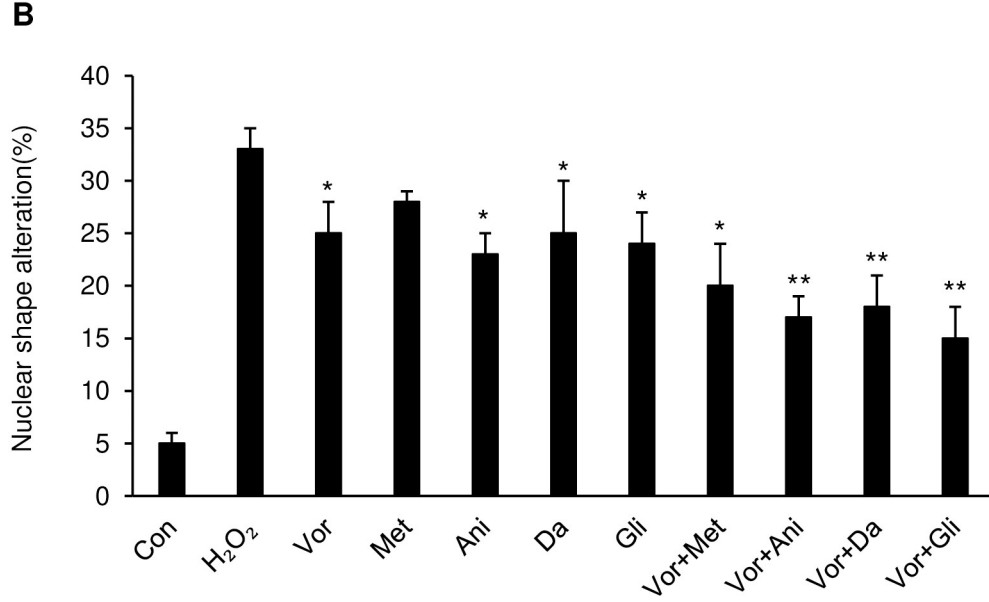

**Fig 5. Investigation of morphologic alterations by nuclear morphology assay.** Nuclear morphology changes in 4′,6-diamidino-2-phenylindole (DAPI)-stained HL-60 cells assessed by fluorescence microscopy. **(A)** HL-60 cells were exposed to $H_2O_2$ (10 μM) for 24 h and then treated with trichostatin A, metformin, danazol, glibenclamide, ampyrone, and each prior drug was co-treated with trichostatin A or **(B)** with vorinostat, metformin, anisomycin, danazol, glibenclamide and each prior drug with vorinostat for 36 h. Nuclear altered shape nuclei were counted and graphed. *p <.05, **p <.01 vs $H_2O_2$ group.

**A**

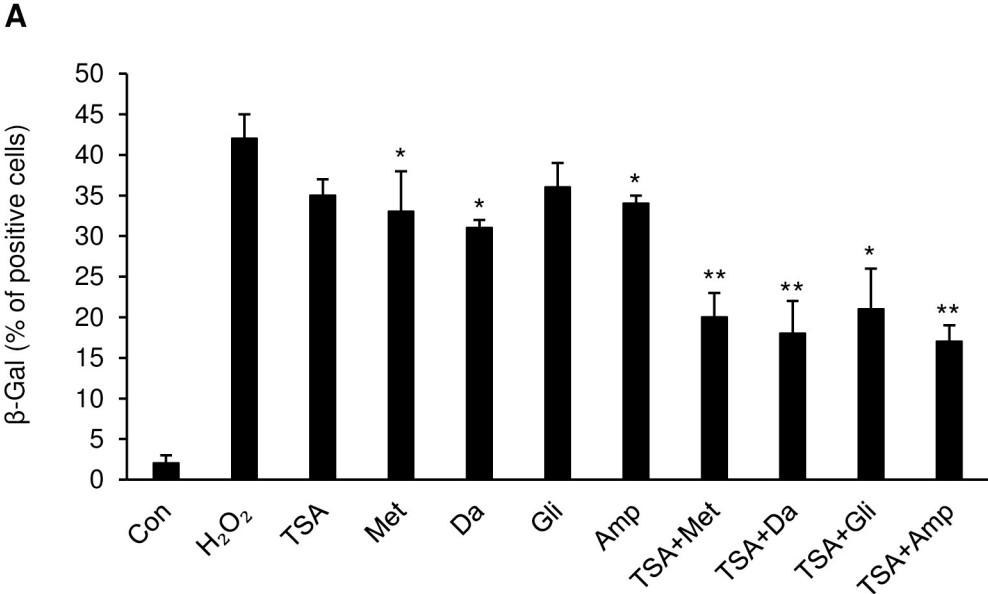

**B**

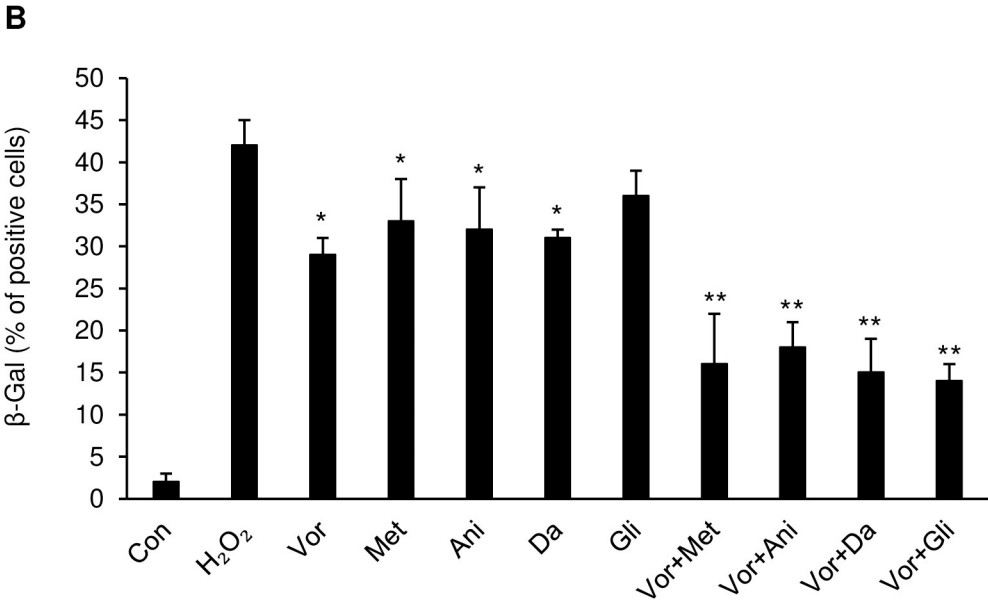

**Fig 6. Combinatorial drug pairs decrease cell senescence in HL-60 cells. (A)** HL-60 cells were exposed to $H_2O_2$ (10 μM) for 24 h and then treated with trichostatin A, metformin, danazol, glibenclamide, ampyrone, and each prior drug was co-treated with trichostatin A. or **(B)** with vorinostat, metformin, anisomycin, danazol, glibenclamide and each prior drug with vorinostat for 36 h. Then the cells were incubated with β-galactosidase and stained cells were counted and plotted. *p <.05, **p <.01 vs $H_2O_2$ group.

models to demonstrate the usefulness of our platform. Although some of these drugs have a potential risk of side effects, they were utilized only as experimental examples in our process.

Trichostatin A serves as an antifungal antibiotic and selectively inhibits the class I and II mammalian histone deacetylase (HDAC) families of enzymes [33]. Vorinostat is also a member of a larger class of compounds that inhibit histone deacetylases (HDAC) [34]. Anisomycin is an antibiotic produced by Streptomyces that inhibits eukaryotic protein synthesis [35]. Thus, these three chemicals are expected to pose nontoxic effects. One of the selected secondary drugs, metformin, is the first-line medication for the treatment of type 2 diabetes [36], particularly in people who are overweight [37]. Metformin is also used in the treatment of polycystic ovary syndrome. Danazol is used in the treatment of endometriosis, fibrocystic breast disease, hereditary angioedema, and other conditions [38, 39]. Glibenclamide is a medication used to treat diabetes mellitus type 2 [40]. Ampyrone is a metabolite of aminopyrine with analgesic, anti-inflammatory, and antipyretic properties [41]. Chlorzoxazone (INN) is a centrally acting muscle relaxant used to treat muscle spasms and the resulting pain or discomfort [42]. Based on their broad indications or combinatory effects (especially for glibenclamide), these drugs may be appropriate as secondary treatments, but not as main compounds for anti-aging treatments.

Interestingly, some of the above agents have been suggested to have anti-aging effects. Danazol may stimulate the recovery of shortened telomeres [43]. Although the exact mechanism has not been revealed, metformin is known to have an anti-aging effect [44]. Glibenclamide induces neurogenesis [45]. Two HDAC inhibitors were suggested as anti-aging agents, which were initially developed as anti-cancer therapeutics. HDAC inhibitors could be a regulator of the aging process due to its epigenetic control ability [46].

Here, we focused on the anti-aging phenotype rather than cell death in leukemia HL-60 cells or cytotoxicity for cancer treatment. Because anti-cancer drugs generally induce cell death and gene expression patterns for cell death can be easily shared [47], they are not suitable for research on gene expression patterns. Thus, we speculated that anti-aging effects could be used as a proper model to study gene expression-based drug repositioning. Leukemia and cancer cells are usually senescence-defective and fail to undergo apoptosis; however; these cells may undergo premature senescence, otherwise called interim cell proliferative arrest, and it has been show that treatments, such as LEE0011, to leukemia cells can induce cell senescence [48–50].

We verified the anti-aging effects of these drugs through different experiments and found that when drugs were mixed, low doses of individual drugs did not influence the overall outcomes. Therefore, our method of identifying combination drug pairs may be sufficient to cover a broad range of disease statuses. This paper showed that CMap-based machine learning for drug repositioning could be used to develop unique targets for drug development through well-designed strategies [51]. In the past, the identification of drugs with drug repositioning has been serendipitous [52, 53]. However, as shown in this paper, rationalized off-target drug repositioning using CMap can take many forms, and modified repositioning involving two-step CMap analyses assisted drug discovery more effectively. It is expected that further bioinformatics analysis, systems biology approaches, or network-based approaches will be applied for new drug discovery as technology develops.

Our computational method to identify combinational drug pairs is useful to cover a broad range of diseases. These results show the potential of our strategy to discover new drug combinations using a large scale input data set that results in a more fine-tuned screening to propose the best anti-aging drug combinations.

## Supporting information

**S1 Table. Architecture of model.**
(DOCX)

**S2 Table. List of datasets used as training dataset.**
(DOCX)

**S3 Table. Performance of the model.**
(DOCX)

**S4 Table. Removed genes using DNNs.**
(DOCX)

**S5 Table. Comparison results of initial matching.**
(DOCX)

**S6 Table. Comparison results of secondary matching.**
(DOCX)

**S7 Table. Neutralized effect of combinatorial drugs.**
(DOCX)

**S1 Fig. Categorization of young and aged groups.** Categorization of patient groups The samples from patients older than 50 were included in the aged group and younger than 30 were included in normal group. The numbers of samples in the normal and aged groups were 84 and 159, respectively.
(PDF)

**S2 Fig. Training of DNN.** Pairs of samples from Control group ('C' in the left side of figure) and Disease group ('D' in the left side of figure) are randomly chosen from the same series in the training dataset and fed to the DNN.
(PDF)

**S3 Fig. Cell viability assay.** (A) trichostatin A (0–300 nM), (B) vorinostat (0–50 μM), (C) anisomycin (0–100 μM), (D) metformin (0–50 mM), (E) glibenclamide (0–100 μM), (F) ampyrone (0–200 μM), (G) danazol (0–200 μM), and (H) chlorzoxazone (0–200 μM) were treated at the different dosages after 3 hours-pretreatment with 10 μM H2O2 to determine the sub-lethal dose (NOAEL, no observed adverse effects level).
(PDF)

**S4 Fig. Cellular ROS levels were attenuated by anisomycin or randomly combined drug pairs.** (A) HL-60 cells were co-treated with anisomycin and metformin or trichostatin A for 36 h after incubation with 10 μM H2O2. (B) Cells were also treated with trichostatin A + chlorzoxazone, vorinostat + trichostatin A, anisomycin + glibenclamide, anisomycin + danazol. The intracellular ROS levels were detected by a microplate reader capable of measuring Ex/Em 495/529 nm spectra and recorded. $^*$p <.05, $^{**}$p <.01 vs H2O2 group.
(PDF)

**S5 Fig. Investigation of nuclear morphology alterations treated with anisomycin or randomly combined drug pairs.** HL-60 cells were treated as indicated in each picture. Nuclei were stained with DAPI, and pictures were taken on a fluorescent inverted microscope. Cells with misshaped(dented) nucleus were indicated by red arrows.
(PDF)

**S6 Fig. Quantification of morphologic alterations treated with anisomycin or randomly combined drug pairs.** Nuclear morphology changes in DAPI-stained HL-60 cells as shown above were counted and graphed. (A) anisomycin and combined drug pairs, (B) trichostatin A + chlorzoxazone, vorinostat + trichostatin A, anisomycin + glibenclamide, anisomycin + danazol. *p <.05, **p <.01 vs H2O2 group.
(PDF)

**S7 Fig. Measuring Senescence by beta-galactosidase staining after treatment with anisomycin or randomly combined drug pairs.** HL-60 cells were treated as indicated in each picture Fluorescent microscopy images were obtained using a fluorescence microscope system, and then beta-galactosidase positive cells were indicated with red arrows.
(PDF)

**S8 Fig. Percentage of beta-galactosidase positive cells by anisomycin or randomly combined drug pairs.** HL-60 cells shown above were quantified and (A) anisomycin and combined drug pairs, (B) trichostatin A + chlorzoxazone, vorinostat + trichostatin A, anisomycin + glibenclamide, anisomycin + danazol. *p <.05, **p <.01 vs H2O2 group.
(PDF)

## Author Contributions

**Conceptualization:** Yi Rang Kim, Jin Woo Choi.

**Data curation:** Sun Kyung Kim, Peter C. Goughnour, Eui Jin Lee, Myeong Hyun Kim, Hee Jin Chae, Gwang Yeul Yun.

**Funding acquisition:** Jin Woo Choi.

**Methodology:** Gwang Yeul Yun, Jin Woo Choi.

**Supervision:** Yi Rang Kim, Jin Woo Choi.

**Writing – original draft:** Sun Kyung Kim, Peter C. Goughnour.

**Writing – review & editing:** Yi Rang Kim, Jin Woo Choi.

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
