## [Decision Letter · Decision Letter 0]

9 Sep 2020

PONE-D-20-24120

Identification of drug combinations on the basis of machine learning to maximize anti-aging effects

PLOS ONE

Dear Dr. Choi,

Thank you for submitting your manuscript to PLOS ONE. After careful consideration, we feel that it has merit but does not fully meet PLOS ONE’s publication criteria as it currently stands. Therefore, we invite you to submit a revised version of the manuscript that addresses the points raised during the review process.

We look forward to receiving your revised manuscript.

Kind regards,

Arun Rishi, Ph.D.

Academic Editor

PLOS ONE

Journal Requirements:

"This research was supported by a grant from the National R&D Program for Cancer

Control, Ministry of Health and Welfare, Republic of Korea (HA17C0039 for YR Kim

and JW Choi). This work was also supported by Basic Science Research Program

through the National Research Foundation of Korea (NRF) funded by the Ministry of

Education, Republic of Korea (2017R1C1B5017615)."

We note that one or more of the authors are employed by a commercial company: name of commercial company.

3.1. Please provide an amended Funding Statement declaring this commercial affiliation, as well as a statement regarding the Role of Funders in your study. If the funding organization did not play a role in the study design, data collection and analysis, decision to publish, or preparation of the manuscript and only provided financial support in the form of authors' salaries and/or research materials, please review your statements relating to the author contributions, and ensure you have specifically and accurately indicated the role(s) that these authors had in your study. You can update author roles in the Author Contributions section of the online submission form.

3.2. Please also provide an updated Competing Interests Statement declaring this commercial affiliation along with any other relevant declarations relating to employment, consultancy, patents, products in development, or marketed products, etc.  

Reviewers' comments:

Reviewer's Responses to Questions

**Comments to the Author**

1. Is the manuscript technically sound, and do the data support the conclusions?

Reviewer #1: Yes

Reviewer #2: Yes

2. Has the statistical analysis been performed appropriately and rigorously? 

Reviewer #1: Yes

Reviewer #2: Yes

3. Have the authors made all data underlying the findings in their manuscript fully available?

Reviewer #1: Yes

Reviewer #2: Yes

4. Is the manuscript presented in an intelligible fashion and written in standard English?

Reviewer #1: Yes

Reviewer #2: Yes

5. Review Comments to the Author

Reviewer #1: Dear Authors,

The topic and the objective of the manuscript very timely and important. The methodology and its validation are interesting, but I have concerns (see below). Generally, I suggest to extend the text of the methods, results and discussion for a better understanding of the broad scientific community. And I also suggest to discuss the novelty of the predictions and the experimental results. In my opinion, the manuscript can be accepted only if the predicted drug combinations have never been tested by other studies.

Line specific comments:

Line 79: duplicated reference ([12] and [15])

Line 94: reference [19] “Mephenytoin: a reappraisal. 1976” doesn’t seem to be an artificial neural network paper.

Line 96: the sentence is about your current work or about [20]? Generally, I suggest not using references within the description of your current results, because it can be confusing.

Line 120: Why did you use a p = 0.1 threshold instead of the regular 0.05?

Line 129: I suggest to put both references into the end of the sentence.

Line 134: You altered the words “aging” / “aged” / “old” for the name of the second group through the manuscript. I suggest using a consistence name of the group, e.g “aged group” everywhere.

Line 192: “and selected 1,415 genes” Totally or this is an additional group?

Line 197: I suggest to replace “S1 Table” by the regular “Supplementary Table S1” and use that format in everywhere.

Line 232: How did you match gene expression data and drugs? I suggest to describe it in the main text too and not only the description of Figure 2.

Line 262: Only 91/617 up-regulated genes matched. Isn’t it a too low proportion?

Line 293: p-value?

Line 302: There is no metformin in Table 1 and Table 2. Why did you test it?

Line 323: I suggest to insert a conclusion here like in the previous section. Are the results support the anti-aging effect of the predicted drug-combinations?

Line 346: Conclusion?

Other comments:

1. Why did you use DNN for determining the differentially expressed genes? It can be calculated directly in more simple ways.

2. Why did you use leukemia study to train the model? Wouldn’t be better a using only healthy young/old samples for studying aging?

3. What about drug-interference? Does exist a gene that is up-regulated by vorinostat and in the same time downregulated by anisomycin? In that case, anti-aging effect of vorinostats would be neutralized by anisomycin.

4. Nobody used these drug-combinations before? Are the experiment are novel results or only reproductions of known effects?

5. Figures are of low quality (hard to read in the PDF). Please use 300dpi.

6. Different doses can resulted in different effect. How do you know which dose should be use? Can the method predict appropriate doses?

Reviewer #2: The paper by Sun Kyung Kim et al. proposed a new machine learning-based strategy to discover more effective anti-aging drug combinations by drug repositioning.

The authors suggested potential drug combinations for anti-aging through the proposed method. In several cell-based assays, they proved the combinational drug treatment had significantly enhanced anti-aging effects in HL60 leukemia cells compared with the condition using a single drug. I think this study can contribute toward developing anti-aging drugs more efficiently by overcoming the limitations of previous drug repositioning strategy.

The manuscript is well written and interesting, but I have some minor questions and suggestions to improve the article before being published.

1. Is there any special reason linked to the aging for using the leukemia patients data and cell line in this study? If so, It would be better to briefly describe about it in the introduction section.

2. Have you checked any molecular senescence-associated markers like p16 in the combi drug treatment condition?

3. Need to improve the resolution of the Figure 1-3.

4. Add the X axis label in the Supple Figure1.

5. What do n-con and con represent in the Figure4B?

Need to unify the label format of the control sample between Fig4A and 4B.

6. PLOS authors have the option to publish the peer review history of their article (what does this mean?). If published, this will include your full peer review and any attached files.

Reviewer #1: No

Reviewer #2: No

---

## [Author Response · Author response to Decision Letter 0]

3 Nov 2020

We appreciate the kind comments from the reviewers and the editorial supports.

---

## [Decision Letter · Decision Letter 1]

11 Dec 2020

PONE-D-20-24120R1

Identification of drug combinations on the basis of machine learning to maximize anti-aging effects

PLOS ONE

Dear Dr. Choi,

Thank you for submitting your manuscript to PLOS ONE. After careful consideration, we feel that it has merit but does not fully meet PLOS ONE’s publication criteria as it currently stands. Therefore, we invite you to submit a revised version of the manuscript that addresses additional minor points indicated by the reviewer 1 below. 

We look forward to receiving your revised manuscript.

Kind regards,

Arun Rishi, Ph.D.

Academic Editor

PLOS ONE

Reviewers' comments:

Reviewer's Responses to Questions

**Comments to the Author**

1. If the authors have adequately addressed your comments raised in a previous round of review and you feel that this manuscript is now acceptable for publication, you may indicate that here to bypass the “Comments to the Author” section, enter your conflict of interest statement in the “Confidential to Editor” section, and submit your "Accept" recommendation.

Reviewer #1: (No Response)

Reviewer #2: All comments have been addressed

2. Is the manuscript technically sound, and do the data support the conclusions?

Reviewer #1: Yes

Reviewer #2: Yes

3. Has the statistical analysis been performed appropriately and rigorously? 

Reviewer #1: Yes

Reviewer #2: Yes

4. Have the authors made all data underlying the findings in their manuscript fully available?

Reviewer #1: Yes

Reviewer #2: Yes

5. Is the manuscript presented in an intelligible fashion and written in standard English?

Reviewer #1: Yes

Reviewer #2: Yes

6. Review Comments to the Author

Reviewer #1: Authors properly answered most of my questions but I still have some minor concerns.

1. The purpose of the developed DNN is still unclear. Authors use the limma package of GEO2R to find DEGs in leukemia using 13 datasets. Then they developed a machine learning model (based on DNN) that virtually do the same: it can find DEGs in leukemia. To me, the difference is that the input of the DNN is two expression values of a particular gene (one from leukemia patients and one from control patients). Authors used DNN for filtering DEGs of the results of the limma package. If I understand correctly, DNN is used to refine the results of limma. Is it true? Overall, I suggest to more written description of the method. In my opinion, in this form, most of the readers (even machine learning and bioinformatics experts) will not understand it. E.g. Please clarify the following questions: what is the exact input of the DNN model? Two real number? What is the exact output? Two real number between 0 and 1? How can you filter by 0.95 if it gives two numbers as an output (Fig S2 shows two output up and down)? Machine learning models usually use test sets and not only training set. Please clarify, why do not you need to use a test set? You trained the model using DEGs in leukemia samples compared to control samples. How can it work for drug-induced data that is based on only leukemia samples?

2. Authors answered to my previous question related DNN as "The original version of the expression profiles was modified using a machine learning system called DNN in order to solve the existing problem of the connectivity map where the ‘inserted gene and the drug-derived gene list’, and the ‘target disease and the experimental cell line’ are different.". It is not clear for me. Why this is a problem? Could you please describe it in more details to readers that are not familiar with CMap (e.g. in the CMap section)?

2. line 91: Citation [18] doesn't fit here (according to the title of the cited paper). Please check in along with all of the citations.

3. line 137: "expressiosn ignatures." Please correct the wrong spelling.

4. line 198: "Cut-off value" of what?

5. Fig1B/Fig3: what are the labels of the x and y axis?

Reviewer #2: In their replies, the authors have satisfactorily addressed the issues raised by reviewers.

I'd like to recommend it for publication in Plos One.

7. PLOS authors have the option to publish the peer review history of their article (what does this mean?). If published, this will include your full peer review and any attached files.

Reviewer #1: No

Reviewer #2: No

---

## [Author Response · Author response to Decision Letter 1]

20 Dec 2020

We greatly appreciate reviewer for the comment.

---

## [Decision Letter · Decision Letter 2]

14 Jan 2021

Identification of drug combinations on the basis of machine learning to maximize anti-aging effects

PONE-D-20-24120R2

Dear Dr. Choi,

We’re pleased to inform you that your manuscript has been judged scientifically suitable for publication and will be formally accepted for publication once it meets all outstanding technical requirements.

Kind regards,

Arun Rishi, Ph.D.

Academic Editor

PLOS ONE

Additional Editor Comments (optional):

Reviewers' comments:

Reviewer's Responses to Questions

**Comments to the Author**

1. If the authors have adequately addressed your comments raised in a previous round of review and you feel that this manuscript is now acceptable for publication, you may indicate that here to bypass the “Comments to the Author” section, enter your conflict of interest statement in the “Confidential to Editor” section, and submit your "Accept" recommendation.

Reviewer #1: All comments have been addressed

2. Is the manuscript technically sound, and do the data support the conclusions?

Reviewer #1: Yes

3. Has the statistical analysis been performed appropriately and rigorously? 

Reviewer #1: Yes

4. Have the authors made all data underlying the findings in their manuscript fully available?

Reviewer #1: Yes

5. Is the manuscript presented in an intelligible fashion and written in standard English?

Reviewer #1: Yes

6. Review Comments to the Author

Reviewer #1: (No Response)

7. PLOS authors have the option to publish the peer review history of their article (what does this mean?). If published, this will include your full peer review and any attached files.

Reviewer #1: No

---

## [Editor Report · Acceptance letter]

20 Jan 2021

PONE-D-20-24120R2 

Identification of drug combinations on the basis of machine learning to maximize anti-aging effects 

Dear Dr. Choi:

I'm pleased to inform you that your manuscript has been deemed suitable for publication in PLOS ONE. Congratulations! Your manuscript is now with our production department. 

Kind regards, 

on behalf of

Prof Arun Rishi 

Academic Editor

PLOS ONE